# Molecular Evaluation of Vitality and Survival Rate of Dormant Kyoho Grape Seedlings: A Step toward Molecular Farming

**DOI:** 10.3390/plants8120577

**Published:** 2019-12-06

**Authors:** Maazullah Nasim, Ting Zheng, Emal Naseri, Xiangpeng Leng, Zhichang Zhang, Haifeng Jia, Jinggui Fang

**Affiliations:** 1Key Laboratory of Genetics and Fruit Development, College of Horticulture, Nanjing Agricultural University, Nanjing 210095, China; Maazullah.nasim@gmail.com (M.N.); 2015204002@njau.edu.cn (T.Z.); 2017104259@njau.edu.cn (E.N.); jiahaifeng@njau.edu.cn (H.J.); 2College of Horticulture, Qingdao Agricultural University, Qingdao, Shandong 266109, China; lengpeng2008@163.com; 3Zhichang Grape Research Institute, Shandong Province, Juxian 276500, China; zhichangnongye@163.com

**Keywords:** seedling vitality, survival rate, molecular evaluation, dormant Kyoho grape

## Abstract

Vitality and survival rate of grape seedlings are crucial factors affecting quality of vineyards. There is no comprehensive study describing accurate evaluation of dormant grapevine seedlings’ vitality and survival rate. The purpose of this study was to explore the possibility of using molecular information to evaluate viability and survival rate of dormant seedlings before transplanting. After bare roots treatment, 1–5 day expression levels of six HKGs in four buds of tetraploid Kyoho grape *(Vitis labruscana: V. labrusca* × *V. vinifera)* seedlings were detected by (Sq.) RT-PCR and qRT-PCR for calibration of the molecular method. The results revealed that HKGs expression indicates vitality and survival of plant, higher expression was strongly linked to higher vitality and survival rate, lower expression was associated with lower vitality, and lowest expression was significantly associated with lowest vitality and survival rate. Moreover, DNA and RNA quality can superficially determine seedling qualities. Finally, the survival rate of the seedlings produced in Juxian-Shandong, Laixi-Shandong, Huailai-Hebei, Suizhong-Liaoning, Changli-Hebei, Guanxian-Shandong, and Zhangjiagang-Jiangsu was 100.00%, 100.00%, 100.00%, 100.00%, 100.00%, 87.77%, and 93.33%, respectively. In conclusion, molecular technique is potential approach for promoting gene information to estimate vitality and survival rate of dormant grape seedlings and might contribute to viticulturists’ efforts.

## 1. Introduction

Grape (*Vitis vinifera L*.) is globally recognized as high-value crop, and table grapes are considered a major horticultural crop of China due to its Chinese origin [1]. Chinese vineyards cover an area of approximately 0.79 million hectares in 2014, and the grape industry has developed robustly in recent past years; China ranks second, contributing 34% to the table grape production globally [2,3,4]. Kyoho (*Vitis labruscana*) grape is a tetraploid and hybrid cultivar *V. labrusca x V. vinifera* [5], which is mainly famous for its large-sized berry, excellent delicious taste, and economically one of the famous grape cultivars in China [6]. This valuable crop suffers severe economic losses by environmental factors, particularly lower temperature during dormancy [7,8]. Therefore, the aforementioned highlighted remarks may convince researchers to turn their research direction toward grapes modification perspectives [9].

Over the last decades, different methodologies have been adopted to evaluate seedling quality [10], which is vital to describe the survival rate extension and predicted to grow and survive successfully after planting [11,12]. The seedling quality assessment can be performed by certain morphological attributes, such as height, stem diameter, root volume [13,14], and physiological characteristics with subsequent field performance after planting which needs long time [15]. Moreover, seedling survival is one of the most critical stages in a plant’s life [16], and environmental stresses can damage the cellular structure and may lead to physiological dysfunctions [17]. Although dormancy is the most tolerant stage of transplanting, long-term dehydration is a challenging stress for dormant grape seedlings [18]. In the nursery, dehydration occurs during and after harvest, which may be the cause of poor growth and sprouting [18,19]. Severe dehydration stress of different tissues in dormancy period can cause seedling death in forthcoming spring [20,21]. Buds are the most sensitive anti dehydration organs [22]. Under the condition of long-term bare root overwintering, buds will gradually dehydrate, resulting in low vitality or death of seedlings [23,24]. Therefore, differences in morphological characteristics often do not reflect variation in physiological conditions [25]. Seedling survival rate is the measuring determiner of how much seedling material in a lot is alive to be reproduced under appropriate field conditions [26], and its viability can be measured by phenological and physiological characteristics, respectively [27]. Many of the tree crops, including grape, have a dormant state, and their seedlings’ survival rate must be determined after growing in a field, which is time-consuming [28,29,30].

Housekeeping genes (HKGs) are typically constitutive genes, known as plant live related genes, that are required for the maintenance of basic cellular functions and are stably expressed in all cells of an organism for cellular survival and other basic cellular functions, including cell wall structure and primary metabolism under normal and pathophysiological conditions [31]. Although some HKGs are expressed at relatively constant rates in most non-pathological situations, the expression of other HKGs may vary depending on experimental conditions. The most suitable HKGs for qRT-PCR in grapevine under abiotic stresses are actin (ACT, GenBank Accession: EC969944), Ubiquitin (UBQ, GenBank Accession: EC929411), glyceraldehyde 3-phosphate dehydrogenase (GAPDH, GenBank Accession: AT1G13440.1) [8], 18S rRNA, and elongation factor1-alpha (EF1α, GenBank Accession: AT5G60390.1) [32,33,34,35]. Moreover, HKGs genes may show uniform expression patterns in various organs of the same plant [36], whereas bud tissue can be the better sample even at the dormant state for gene expression analysis [37].

To our knowledge, no attempt has yet been made to measure the vitality and survival rate of grape seedlings at dormant state on molecular basis. Therefore, the present study was conducted to evaluate the vitality and survival rate of dormant Kyoho grape seedlings via molecular approach obtained from 07 geographical regions of China. The present study may contribute to the early assessment of seedling vitality and survival rate at the dormant stage before field plantation, which was preciously executing by long time growing trial.

## 2. Results

### 2.1. Calibration of Dormant Seedlings’ Vitality and Survival Rate Evaluation by Molecular Techniques

#### 2.1.1. Microscopic View of Buds and Stem Segments of Treated Seedlings

To check the seedlings viability and dehydration, bud and stem segments of the seedlings were observed by stereoscopic microscope to identify the differences in viability and dehydration. In the control, all the buds and stem segments looked green and were in better status for growing in the field. After one day of treatment, only the 4th bud (upper bud) and stem segment seemed dehydrated; its color was changed from green to brownish compared to other buds. After two days of treatment, the 3rd and 4th buds and stem segments seemed dehydrated and died, whereas the two lower parts were less affected. Three days, the 2nd, 3rd, and 4th buds and stem segments were dried and dehydrated, whereas the 1st parts were appeared fewer dehydrated. After four days, all the four parts of seedlings were dehydrated, the color of the upper parts was darker while it was light in the lower parts, some cracks were also seen in the upper parts of the stem. Finally, after five days, all the buds and stem segments were completely dehydrated and dried as shown in Figure 1.

#### 2.1.2. DNA and RNA Quantification and Quality Analysis of the Treated Seedlings

Quality and quantity of DNA and RNA extracted from the four parts of dormant grape seedlings treated 0–5 days of bare roots were examined. In the control group, the quality of DNA and RNA in all buds’ tissues was great in size and shape, and its optical visualization was similar to each other. After one day of treatment, 1st, 2nd, and 3rd buds’ DNA and RNA integrity were better, the main bands were clear and milky white, OD_260_/_280_, _260_/_230_, and concentration was higher. However, 4th bud’s main bands were darker; OD_260_/_280_ was 1.18, 1.19 and OD_260_/_230_ was 1.03, 0.93, which indicated poor quality of DNA and RNA. After two days treatment, the 1st and 2nd bud’s integrity, purity, and optical density revealed better quality and its color was white, but the 3rd and 4th bud’s DNA and RNA quality was poor, OD was lower, and color of the main bands were darker. After three days treatment, 1st bud’s purity and quality showed better condition of the bands, whereas the 2nd, 3rd, and 4th buds revealed more degradation and low concentration; thus, their bands were also not clear. After four- and five-day treatments, the DNA and RNA quality, purity, and integrity of all four buds intimated poor quality, bands were dark, and OD and concentrations were lower in all the buds. Thus, DNA and RNA showed poor quality as shown in Table 1 and Table 2 and Figure 2 and Figure 3. Additionally, the results indicated that DNA and RNA quality is associated with the seedling quality. Therefore, the presence and absence of DNA and RNA and its quality and quantity measurement can be used as a new criterion to assess quality of dormant grape seedlings superficially prior planting in the field.

#### 2.1.3. Expression Analysis of HKGs for Evaluation of Seedlings Vitality and Survival Rate 

The expression of six HKGs in four buds of dormant grape seedlings during their declining vitality and survival was analyzed by (Sq.) RT-PCR and qRT-PCR. The results exhibited differences in the expression level of the HKGs, different lower expression trends were observed while the vitality was decreased from upper to lower part of the seedlings respectively. Moreover, higher expression of HKGs showed higher survival of the seedlings. The results also showed that the expression level of HKGs was closely related to the vigor and viability of grape seedlings, which might be used to diagnose vitality and survival rate of grape seedlings.

The expression level of Actin, GAPDH, UBQ, EF1r, Tubulin, and 18s rRNA genes in 1st bud determined by (Sq.) RT-PCR and qRT-PCR, shown in Figure 4 and Figure 5, were higher in control, which showed higher vitality and survival rate. After one day treatment, the expression level of HKGs was high which presented higher survival rate and low vitality, and the expression level was significantly (*p* < 0.05) decreased gradually after two and three days of treatment which showed lower vitality and survival rate. However, after four days of continuous treatment, the expression was obviously decreased and no expression was observed after five-day treatment and seedlings were not survived as result.

Expression level of HKGs in 2nd bud showed in Figure 6 and Figure 7 was significantly (*p* < 0.05) decreased gradually after one and two days treated of bare roots condition which showed lower vitality of buds, and clearly significant (*p* < 0.05) decreased after three-days of treatment which exposed lower survival of the seedlings. However, lowest expression was seen after four days of treatment, which indicated plant death, whereas no expressions were detected after five-day treatment, consequently.

In the 3rd bud, the expression level of HKGs gradually decreased after one day treated of bare roots condition which revealed low vitality, and significantly (*p* < 0.05) decreased after two and three days of treatment showed lower vigor and lower vitality. However, no survival was seen in the buds while there was no expression of HKGs after treatment of four and five days as shown in Figure 8 & Figure 9.

In the 4th bud, the expression level of HKGs obviously significant (*p* < 0.05) decreased after one day treated of bare roots condition resulted in death of the buds. No expression was seen after three to five days of treatment. Moreover, the seedlings survival decreased with increasing number of bud death from upper to lower part respectively as shown in Figure 10 and Figure 11. The expression level of HKGs detected by (Sq.) RT-PCR can rapidly and accurately determine vitality and survival rate of the seedlings at dormant state, whereas qRT-PCR can give us more accurate and reliable data. 

#### 2.1.4. Verification of Molecular Evaluation of Treated Seedlings Vitality and Survival Rate Using Field Data

Growth parameters were observed at 10, 20, and 30 Days after First Bud Burst (DAFBB). The results revealed that bud survival rate, shoots length, and number of leaves per shoot were gradually decreased and bud burst time was delayed from 1–5 days, after bare roots treatment from upper to lower part, respectively. After one day treatment, no bud burst was seen in the upper part, and, although the other lower buds survived, shoot length and number of leaves per shoot decreased compared to control from upper to lower parts of the seedlings. After two days of treatment, 1st and 2nd bud showed growth but no bud burst was seen in 3rd and 4th buds, indicating that dehydrated buds showed no growth. After three days of treatment, three upper buds of the seedlings died while the lower bud was survived. However, bud burst was delayed, shoot length was shorter and number of leaves was fewer compared to control group. Finally, after 4–5 days treatment, all the buds died and no bud burst was seen, representing that the seedlings were completely died and viability was not observed. Seedling vitality (shoot length and number of leaves per shoot) decreased with increasing number of died buds from upper to lower part of the seedlings. Finally, the survival rate on daily base treatment from 0 to 5 was 100.00%, 85.55%, 75.55%, 51.11%, 2.22%, and 0.00%, respectively, as shown in Figure 12 and Table 3.

#### 2.1.5. Strategy to Pre-Evaluate Survival of Dormant Seedlings Early in Winter

Higher expression of HKGs exposed higher vitality and survival rate; high expression revealed low vitality and high survival rate, whereas its lower expression indicated lower vitality and survival rate. In addition, death of all buds in the seedlings exposed no expression of HKGs (Table 3 and Figure 4, Figure 6, Figure 8, and Figure 10). However, seedlings quality can be roughly evaluated by DNA and RNA quality and quantity analysis (Figure 2 and Figure 3). We randomly evaluate the four buds of purchased seedlings by qRT-PCR, it can accurately determine possible growth level and survival rate at dormant stage, the seedlings with 4, 3, 2, and 1 alive buds indicated 100%, 85.55%, 75.55%, and 51.11% survival rate, respectively. Meanwhile, the bud burst time was later and shoot length was shorter in the seedlings with low survival rate from upper to lower part respectively (Table 3). To conclude, qRT-PCR technique is accurate and rapidly applicable for evaluation of the seedlings’ vitality and survival rate at dormant stage prior planting in the field and could avoid time consuming. This technique will open a new window for horticulturists to evaluate all deciduous fruits’ seedlings vitality and survival rate at dormant state.

### 2.2. Practical Usage of Molecular Techniques to Evaluate Vitality and Survival Rate Differences of Dormant Grape Seedlings Produced in Different Areas

#### 2.2.1. DNA and RNA Quantification and Quality Analysis

DNA and RNA integrity were assessed extracted from four buds of dormant grape seedling tissues within three replicates produced in seven different geographical regions, 1% agarose gel electrophoresis patterns showed that bands of the buds’ tissues produced in ZJ had lower quality, less brightness, and some dispersion phenomena with the partial decline of viability of seedlings; DNA and RNA similarly seemed to be partially damaged compared to seedlings produced in GS. Quality of seedlings produced in GS was lower according to OD_260_/OD_280_ and OD_260_/OD_230_ compared to SL, CH, and HH seedling tissues. Bands of the bud tissues produced in JS and LS were clearer, its OD_260_/OD_280_, OD_260_/OD_230_, and integrity were best compared with the seedlings produced in SL, CH, and HH, respectively as shown in Table 4 and Table 5 and Figure 13 and Figure 14.

#### 2.2.2. Pre-Verification of Seedling Vitality and Survival Rate by qRT-PCR

We analyzed the cDNA tissue samples of four buds of Kyoho dormant grape seedlings, which were produced under normal conditions in seven different geographical regions of China, we also analyzed the expression level of 6HKG (Actin, GAPDH, UBQ, EF1r, Tubulin, and 18srRNA). The results showed that the expression level of HKGs in 1st bud of seedlings produced in SL, CH, HH, JS, and LS was significantly higher compared to ZJ and GS indicated higher survival rate of the seedlings. The expression level of 1st, 2nd, and 3rd buds of the seedlings produced in JS, LS was higher compared to SL, CH, and HH seedlings indicated higher vitality (shoot length and number of leaves per shoots). The expression level of HKGs in the 3rd and 4th bud of ZJ, GS was significantly (*p* < 0.05) lower compared to seedlings produced in SL, CH, and HH indicated low vitality and low survival rate of the seedlings. Expression level of HKGs in 3rd and 4th buds of GS was significantly (*p* < 0.05) lower compared to SL, CH, and HH, indicated lower survival rate and lower vitality. The lowest significant (*p* < 0.05) level of HKGs expression was observed in 4th bud of seedlings produced in ZJ, which indicated the lowest survival rate and lowest vitality among all as shown in Figure 15 and Figure 16.

To conclude, higher expression of HKGs in four buds of dormant grape seedlings showed higher survival rate and higher vitality, as indicated in JS and LS regions seedlings (100%); higher expression of 1st bud; and high expression of 2nd, 3rd, and 4th buds, indicating higher survival rate and low vitality as specified in the seedlings produced in SL, CH and HH, low expression of 1st bud and lower expression of 2nd, 3rd, and 4th bud, respectively, designated lower vitality and lower survival rate as per seedlings produced in GS and ZJs’ seedlings exposed 93.33% and 87.77% survival rate respectively.

#### 2.2.3. Verification of the Accuracy of Pre-analyzed Seedling Survival Rate Using Field Data 

To confirm molecular evaluation of the seedlings’ vitality and survival rate, the growth performance of four buds (shoot length and number of leaves per shoot) in the seedlings was observed on days 10, 20, and 30, after first bud burst (FBB). First bud shoot length and its number of leaves in the seedlings produced in SL, CH, HH, JS, and LS were significantly higher compared to the 1st bud of seedlings produced in ZJ and GS, survival rate of the seedlings was also high in SL, CH, HH, JS, and LS compared to seedlings produced in ZJ and GS. Growth performance of the 2nd, 3rd, and 4th buds of the seedlings produced in JS, LS were significantly higher compared to SL, CH, and HH, which exhibit higher vitality as it was indicated in molecular examinations. The growth and survival rate of the 2nd, 3rd, and 4th buds in the seedlings produced in GS were significantly lower compared to the seedlings produced in SL, CH, and HH. Finally, the lowest significant growth and survival rate of the seedlings were observed according to field data in the seedlings produced in ZJ compared to all. The survival rate of the seedlings produced in SL, CH, HH, JS, LS, ZJ, and GS were 100%, 100%, 100%%, 100%%, 100%%, 87.77%, and 93.33%, respectively, as presented in Figure 17 and Table 6. Remarkably, the growth performance showed similar results to molecular evaluation and confirmed accuracy of the techniques which can rapidly and accurately evaluate vitality and survival rate of the grape seedlings at dormant stage prior planting in the field.

## 3. Discussion

Modern molecular biology provides researchers with abundant HKGs information; however, access to this information for use as internal reference genes for quantitative analysis of gene expression is limited. Therefore, the novelty of this research is application of HKGs information for estimation of vitality and survival rate of grape seedlings at dormant state. Previously, survival rate and vitality of deciduous fruits seedlings were evaluated by long time growing trial; we applied molecular techniques to estimate dormant grape seedling vitality and survival rate rapidly and accurately.

DNA-based diagnostic detection provides an ideal method for detecting infections rate in public health [38,39]. DNA exists relatively after cell death, but its quality is lower than that of living cells [40]. In addition, gel electrophoresis is one of the standard methods to separate and identify the purity of DNA fragments, which is a simple and fast technique which can distinguish DNA fragments from mixtures [41,42]. In this study, the seedlings tissues that were alive and healthy showed milky white color and no degradation of the bands were observed. After bare roots treatment, the quality and integrity of DNA, as well as its viability, were gradually decreased; DNA was severely degraded with the decreasing of seedling survival, causing irreversible damage in the DNA (Table 1 and Figure 2). In addition, the DNA of four buds of dormant grape seedlings produced in different Chinese geographical regions exposed a clear DNA band, high integrity without dispersion, and degradation, indicating better quality compared to the seedlings with low quality (Table 4 and Figure 13). 

RNA quality is determined by its purity and integrity [43], which are important for meaningful downstream experiments [44]. As a nucleic acid, RNA is widely used for protein synthesis and gene expression patterns in different plants [44]. High-quality RNA will show a typical expression, whereas low-quality RNA affects PCR amplification and influences the reliability of qRT-PCR [45]. It is well known that the quantity and quality of RNA affects gene expression by qRT-PCR, and low-quality RNA affects the results of downstream applications [46]. Moreover, low-quality RNA may be due to cell degradation, cell death, dehydration, and/or cell structure damage [47,48,49]. High integrity RNA and cDNA synthesis will show normal expression of HKGs genes in qRT-PCR [50]. In the present study, the seedlings tissues that were alive and healthy showed clear two bands of RNA, milky-white color, and no contamination and degradation of the bands was observed. After bare roots treatment, the quality and integrity of RNA as well as the vitality and survival rate of seedlings gradually decreased in different buds of seedlings in a time-based manner from upper to lower parts, respectively (Table 2 and Figure 3). In addition, four bud tissues of dormant grape seedlings produced in different areas exposed two clear bands (18S and 28S) of RNA with high integrity without dispersion and degradation, which displayed high percentage of survival rate in contrast to the low survival rate of the seedlings after planting in the field (Table 5 and Figure 14). It is anticipated that DNA and RNA quality assurance technologies will be productive benchmark method for the determination of seedling viability and survival rate measurement in near future, and will be used for scientific research and arbitration in production disputes over seedling quality.

At present, physiological indicators are using enormously for seedling quality evaluation, which required complex and expensive equipment, time-consuming, and most of them need active growth of seedlings [51,52,53,54]. RT-PCR can detect the presence of nucleic acids and enumerate bacterial strains [55,56]. Molecular techniques such as PCR, qPCR, and viability PCR are widely used in public health to diagnose virus infection and detect micro-organisms [57,58]. Detection of gene expression is an important tool to study biological processes [50]. HKGs are most stable genes and are important for cellular survival and basic cellular functions, and may show stable expression during the treatments in molecular biology researches in alive plants [50,59,60]. qPCR method can differentiate between viable and nonviable bacteria in environmental water samples [61]. Rapid viability detection of *Bacillus anthracis* (spore-forming, Gram-positive bacterium; the causative agent of the zoonotic disease) can be performed by qPCR [62,63,64]. In this study, we used HKGs expression to estimate vitality and survival rate of the dormant grape seedlings; the technique was more accurate, rapid, and applicable in the dormant state. Normal expression of housekeeping genes in four buds were observed in vigorous dormant seedlings, and, with decreasing viability, gene expression decreased gradually from upper to lower parts, respectively, whereas there were no detection and expression of HKGs in death parts of the seedlings (Figure 4, Figure 6, Figure 8, and Figure 10). Moreover, different expression patterns of HKGs in four buds of the seedlings produced in different areas indicated different survival rate and vitality, the technique is more suitable and applicable in dormant state (Figure 15 and Figure 16). 

Seedling quality and morphological growth can be measured on the base of shoot growth and length [13,14,15,65]. One of the main reasons for poor germination and establishment of bare-root deciduous plants is the dehydration pressure during and post-harvest in the nursery, and dehydration may occur at other times before planting [66,67,68]. Bud is the most appropriate organ to determine the dehydration status of seedlings, and the hydration status during dormancy has a significant correlation with the growth potential and survival of plants [21]. Dehydration tolerance in dormant grape seedlings is associated to root size [67]. We confirmed our pre evaluation of the treated seedlings vitality and survival rate using growth data, similarity of growth data to molecular evaluation indicated strength and accuracy of our methods (Table 3; Figure 1 and Figure 12). Differences of vitality and survival rate of the seedlings produced in different regions were evaluated by growth and morphological specifications after planting to approve the strength of molecular evaluation (Table 6; Figure 17). The aforementioned results indicate that molecular techniques were accurate and applicable methods to measure the vitality and survival rate of dormant grape seedlings rapidly and accurately prior planting in the field.

## 4. Materials and Methods

### 4.1. Plant Materials and Growth Condition

#### 4.1.1. Experimental Design for Calibration of Seedlings’ Molecular Evaluation

This experiment was conducted in white horse research center of Nanjing agricultural University, Nanjing China. Seven-hundred-and-thirty-eight one-year-old healthy dormant Kyoho grape seedlings produced from cutting, produced in Laixi district of Shandong province, were randomly divided into six groups using the method of completely randomized design [69]. Each group covers 123 seedlings. The aforementioned groups consisted of Control, 1-day, 2-days, 3-day, 4-day, and 5-day, each group had 3 replicates, and each replicate contained 30 seedlings for planting and 11 for sampling. The upper parts of the seedlings (after 4th bud from ground level) were cut for equal results of vitality and survival rate comparison prior to commencing our respective research treatment. All seedlings were treated in bare roots inside the control greenhouse under 11L: 13D photoperiod, 31% average relative humidity and greenhouse temperature were adjusted in 27 °C: 22 °C day/night for 5 consecutive days. After ending each day treatments, sampling purpose seedlings were cut into four parts (Figure 18). Planting purpose seedlings were planted in the pots for evaluating vitality and survival rate, the samples were collected from four buds of seedlings for RNA quality analysis, DNA quality analysis, and genes expression from 26 February to 03 March, 2019. 

#### 4.1.2. Experimental Design for Practical Usage of Molecular Evaluation to Estimate Dormant Seedlings Vitality and Survival Rate

One-year-old healthy Kyoho grape seedlings with same morphological specifications produced in seven different geographical regions of China were selected for the experiment. These regions are as follows. 1: Suizhong district of Liaoning Province (SL), 119.699379N, 40.37773E, 180EL. 2: Changli district of Hebei Province (CH), 119.219251N, 39.708778E, 72 EL. 3: Huailai district of Hebei Province (HH), 115.576165N, 40.434583E, 144EL. 4: Juxian district of Shandong Province (JS), 118.739449N, 35.563684E, 403EL. 5: Laixi district of Shandong Province (LS), 120.38459N, 36.901626E, 177EL. 6: Zhangjiagang district of Jiangsu Province (ZJ), 120.560644N, 31.917206E, 160EL. 7: Guanxian district of Shandong Province (GS), 115.52988N, 36.512144E, 660EL. The aforementioned seedlings were used to determine the vitality and survival rate differences among them on molecular base. A total of 903 seedlings were used in the experiment; however, for the determination of survival rate, we selected 129 seedlings from each region, each region’s seedlings had 3 replicates, 3 × 13 = 39, for sampling, and (3 × 30 = 90) seedlings for planting purpose, and the same grouping system was applied for the rest of the relevant regions’ seedlings. Similarly, the upper parts of the seedlings (25 mm above from 4th bud using ordinary ruler commencing ground level) were cut in a horizontal direction by pruning scissor for growth comparison, the bud samples were collected from four parts of dormant seedlings, as shown in Figure 18, and then immediately frozen in liquid nitrogen and stored in −80 °C [42]. Finally, planting purpose seedlings were planted in 46% peat soil + 27% pearlite + 27% vermiculite substrate inside the control greenhouse under 11L: 13D photoperiod and 31% average relative humidity, and greenhouse temperature was adjusted in 27 °C: 22 °C day/night on 20^th^ of February, 2019. Regularly Irrigation was performed according environmental conditions.

### 4.2. DNA Isolation and Quality Analysis

Genomic DNA was isolated according to the protocol of Vazyme FastPure Plant DNA Isolation Mini Kit (Vazyme Biotech Co., Ltd., Nanjing, China). Twenty milligram bud samples were scaled accurately and ground immediately in liquid nitrogen, we then added 400 µL of Buffer A1 and 4 µL of RNase A (10 mg/mL). Later, we vortex-mixed thoroughly and gave 65 °C water bath for 10 min. The tubes were inverted 2–3 times during the water bath process to mix the samples accurately, we then added 130 µL Buffer A2 to the mixture and placed on ice for 5 min. These tubes were centrifuged at 14,000 rpm (−18,400× *g*) for 10 min and transferred its supernatant to new tubes. In accordance to the protocol, we added 1.5 times the supernatant volume of Buffer A3 and mixed immediately, the mixture (including the precipitate) was transferred to FastPure gDNA Columns IV (the adsorption column was placed in the collection tube) and centrifuged at 12,000 rpm (−13,400× *g*) for 1 min. After the filtrate were discarded, we added 600 µL of Buffer AW, centrifuged at 12,000 rpm (−13,400× *g*) for 30 sec, discarded the filtrate (this step was repeated two times), and then placed the adsorption column back into the collection tubes and centrifuged at 12,000 rpm (−13,400× *g*) for 2 min to remove the rinse solution and prevent residual ethanol from inhibiting the downstream reaction. Last, we placed the column into a new tube and added 50 µL preheated 65 °C elution buffer to center of the adsorption column and stranded at room temperature for 3–5 min, then centrifuged at 12,000 rpm (−13,400× *g*) for 1 min. The extracted DNA was stored in −20 °C for downstream experiments.

Quantification of DNA was done using spectrophotometric measurement of UV absorption at wavelengths 230, 260, and 280 nm. DNA purity was determined by the optical density (OD) Values ratio OD_260_: OD_280_ and OD_260_: OD_230_. These ratios give the indication of contamination of protein, polyphenol, and carbohydrate respectively [42]. The DNA concentration was calculated using formula DNA concentration (µg/mL) = OD described in [42].

### 4.3. RNA Isolation and Quality Analysis 

Total RNA was isolated according to the protocol of the E.Z.N.A. ^®^Plant RNA Kit. No. R6827-02. Kit (OMEGA bio-tek Co., Ltd., Guangzhou, China). Fifty milligram bud samples were separately ground in liquid nitrogen and immediately added 500 μL RCL Buffer into the tube, then vortex at maximum speed to mix thoroughly and then incubated at 55 °C for 3 min. We centrifuged these samples at 10,000× *g* for 5 min at room temperature, cleared lysate was transferred into a gDNA Filter Column in 2 mL collection tube and centrifuged at 14,000× *g* for 2 min at room temperature, according to the protocol, then we added 1 volume RCB Buffer to the flow-through and vortex at maximum speed for 20 s. From the achieved mixture, we transferred 700 µL samples to HiBind^®^ RNA Mini Column and centrifuged at 12,000× *g* for 1 min, added 400 μL RWF Wash Buffer, and centrifuged at 10,000× *g* for 30 s. Later, we added 500 μL RNA Wash Buffer II and centrifuged at 10,000× *g* for 30 s (repeated two times) and the filtrated residue was discarded. The collection tube was reused and centrifuged at maximum speed for 1 min to completely dry the HiBind^®^ RNA Mini Column, Transferred the HiBind^®^ RNA Mini Column to a clean 1.5 mL micro centrifuge tube. Finally, we added 50 μL 65 °C preheated DEPC water, centrifuged at maximum speed for 1 min and stored eluted RNA at −70 °C up to downstream experiments.

Quantification of RNA was done by spectrophotometric measurement of UV absorption at wavelengths 230, 260 and 280 nm. RNA integrity and degradation were determined by OD Values ratio OD_260_: OD_280_ and OD_260_: OD_230_. RNA quality was checked in 1% Agarose gel electrophoresis, the voltage was about 150 V. the photograph was taken out under the UV imaging system on Bio photometer. An equal amount of RNA was used to differentiate the quality. 

### 4.4. cDNA Synthesis

cDNA was synthesized according to the protocol of 5X All-In-One MasterMix (with AccuRT Genomic DNA Removal) kit (abmGood Co., Ltd.). RNA Template 2 μg, AccuRT Reaction Mix (4X) 2 μL, and Nuclease-free H_2_O 4 μL were mixed, the mixture was Incubated at 42 °C for 2 min. Next, we added AccuRT Reaction Stopper (5X) 2 μL and mixed thoroughly as well as 5X All-In-One MasterMix 4 μL and Nuclease-free H_2_O 6 μL until the total reaction volume was 20 μL. The mixture was incubated at 25 °C for 10 min, then at 42 °C for 15 min. The reaction was inactivated at 85 °C for 5 min, followed by 4 °C for 2 min. Quality of cDNA was checked by spectrophotometer.

### 4.5. (Sq.) RT-PCR Analysis

(Sq.) RT-PCR was performed by 2 x Hieff^®^ PCR MasterMix with Dye. CAT: 10102ES03. The mixture was 1 μL cDNA, 1 μL forward primer, 1 μL reverse primer, 12.5 μL 2 x Hieff^®^ PCR Master Mix (with Dye), and 9.5 μL DEPC water in a total volume of 25 μL. The mixture was Incubated at 94 °C for 10 min, 94 °C for 30 s, 58 °C for 30 s, and 72 °C for 1 min, followed by 72 °C for 8 min and 16 °C for 2 min; steps 2, 3, and 4 were repeated as follows; 18, 21, 23, 27, and 30 cycles were used for Actin, GAPDH, UBQ, and EF1r; 27, 30 33, 36, and 39 cycles were used for Tubulin; and 12, 15, 18, 21, and 23 cycles were used for 18S rRNA. 

### 4.6. qRT-PCR Analysis

To evaluate expression level of selected 6 HKGs, quantitative real-time polymerase chain reaction (qRT-PCR) was performed in three replicates using the cDNA samples. The qRT-PCR reaction comprised 5 μL of Hieff ^TM^ qPCR SYBR^®^ Green Master Mix. CAT: 11202ES03. (YEASEN), 0.4 μL of forward primer, 0.4 μL of reverse primer, 1 μL of cDNA, and 3.2 μL of RNase-free water in a total volume of 10 μL. The whole processes were conducted in a Quantagene q225 Real-Time PCR system (Kubo Tech Co., Ltd. Tianjin, China) Each gene was replicated 3 times in qRT-PCR assay, and relative gene expression was calculated using the CT value method [70]. The primers used for (Sq.) RT-PCR and qRT-PCR are listed in Table 7.

### 4.7. Statistical Analysis

The statistical analysis were calculated by IBM SPSS statistics, version 19 software (SPSS Inc., Chicago, IL, USA), and graphs were constructed using GraphPad Prism 6 software (IBMP Crop, United states of America). Significant differences were analyzed by one-way analysis of variance (ANOVA) followed by the Duncan post hoc test. Gene expression profiling was calculated using the CT value method (*n* = 3) [70]; DNA and RNA quantification data are presented as mean and standard error (SE) of means (*n* = 3), and field data (seedlings growth) are presented as mean and standard division of means (*n* = 30). Different letters (marked a-g) indicated statistical differences at *p* < 0.05 determined by Duncan’s multiple range tests as shown in respective figures and tables. 

## 5. Conclusions

In this research, we applied molecular techniques to estimate dormant seedlings vitality and survival rate prior planting in the field. We found that qRT-PCR is potential applicable determiner for the vitality and survival rate at dormant state. To conclude, DNA and RNA quality analysis can roughly evaluate quality of the dormant seedlings. However, HKGs are considerable genes for determining the seedlings vitality and survival rate. The survival rate of many deciduous fruits including grapes can be estimate by applying molecular techniques using HKGs expressions at dormant stage. Therefore, these insights may help professionals to select the best source of seedlings for establishment of deciduous fruits gardens. 

## Figures and Tables

**Figure 1 plants-08-00577-f001:**
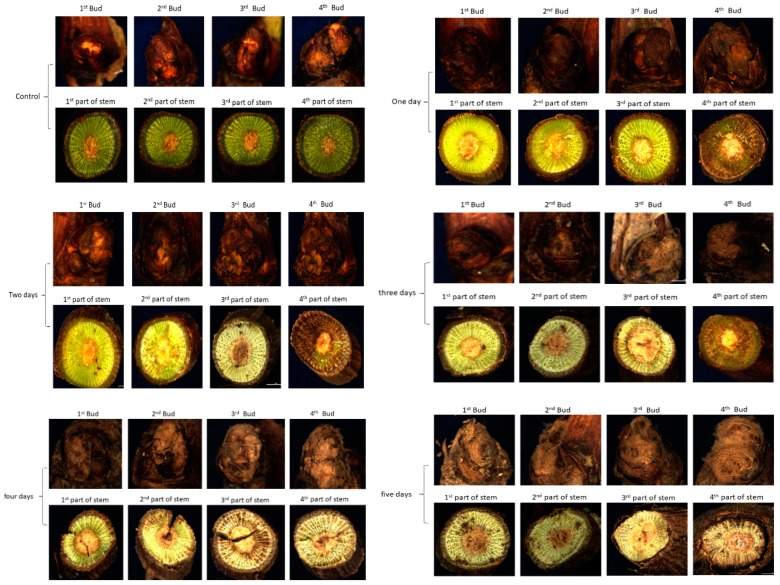
Microscopic appearance of bud and stem segments of dormant grape seedlings after 0–5-day bare roots treatment.

**Figure 2 plants-08-00577-f002:**
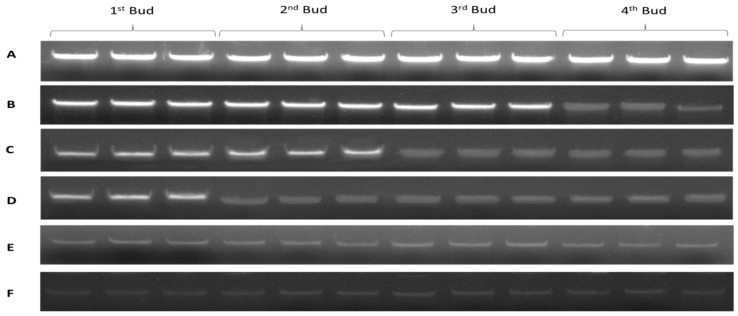
Agarose gel analysis of genomic DNA extracted from four buds of dormant grape seedlings. (**A**) Control, (**B**) 1 day, (**C**) 2 days, (**D**) 3 days, (**E**) 4 days, (**F**) 5 days of bare roots treatment.

**Figure 3 plants-08-00577-f003:**
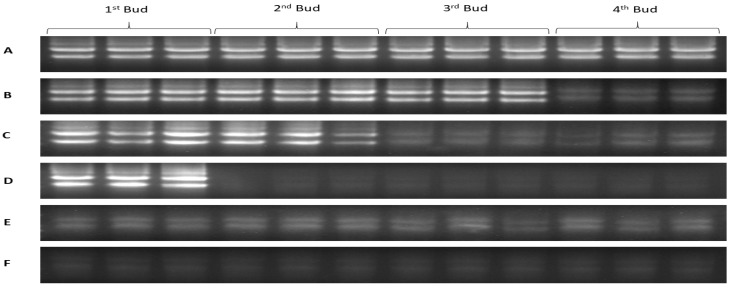
Agarose gel analysis of total RNA extracted from four buds of dormant grape seedlings. (**A**) Control, (**B**) 1 day, (**C**) 2 days, (**D**) 3 days, (**E**) 4 days, (**F**) 5 days.

**Figure 4 plants-08-00577-f004:**
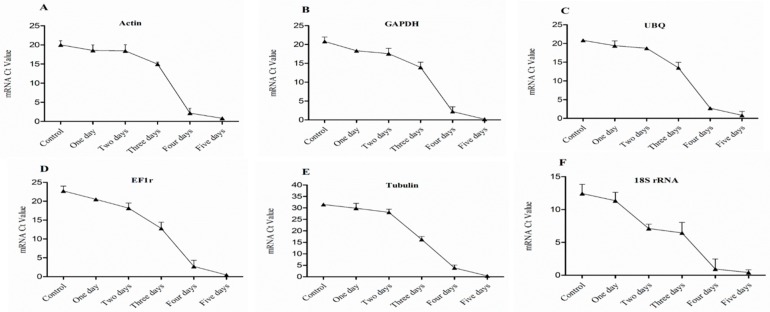
Expression level of mRNA in 1st bud (base bud) of dormant grape seedlings. (**A**) Actin, (**B**) GAPDH, (**C**) UBQ, (**D**) EF1r, (**E**) Tubulin, and (**F**) 18S rRNA were determined by qRT-PCR (1–5 days). Vertical bars represented standard error (SE) of means (*n* = 3). Different values indicated as statistical differences at *p* < 0.05 as determined by Duncan’s multiple range tests.

**Figure 5 plants-08-00577-f005:**
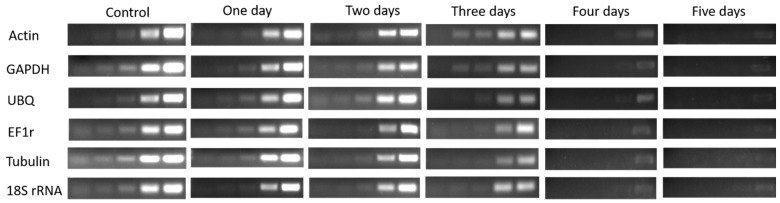
(Sq.) RT-PCR analysis of mRNA level in the 1st bud during 1–5 days of treatment in dormant grape seedlings. 18, 21, 23, 27, and 30 cycles were used for Actin, GHAPDH, UBQ, and EF1r; 27, 30 33, 36, and 39 cycles were used for Tubulin; and 12, 15, 18, 21, and 27 cycles were used for 18S rRNA.

**Figure 6 plants-08-00577-f006:**
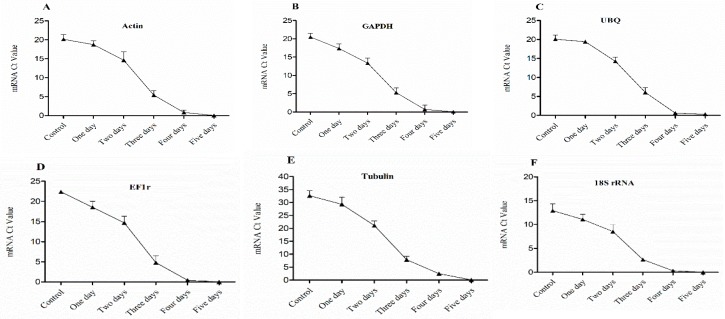
Expression level of mRNA in 2nd bud of dormant grape seedlings, (**A**) Actin, (**B**) GAPDH, (**C**) UBQ, (**D**) EF1r, (**E**) Tubulin, and (**F**) 18S rRNA were determined by qRT-PCR (1–5 days). Vertical bars represented standard error (SE) of means (*n* = 3). Different values indicated as statistical differences at *p* < 0.05 as determined by Duncan’s multiple range tests.

**Figure 7 plants-08-00577-f007:**
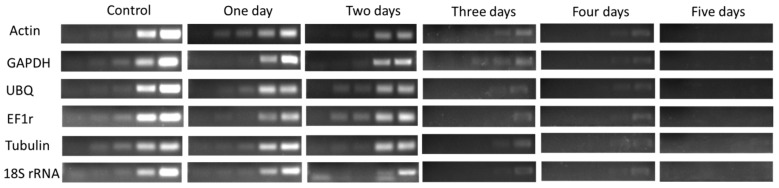
(Sq.) RT-PCR analysis of the mRNA level in the 2nd bud during 1–5 days of treatment in dormant grape seedlings. 18, 21, 23, 27, and 30 cycles were used for Actin, GHAPDH, UBQ, and EF1r; 27, 30 33, 36, and 39 cycles were used for Tubulin; and 12, 15, 18, 21, and 27 cycles were used for 18S rRNA.

**Figure 8 plants-08-00577-f008:**
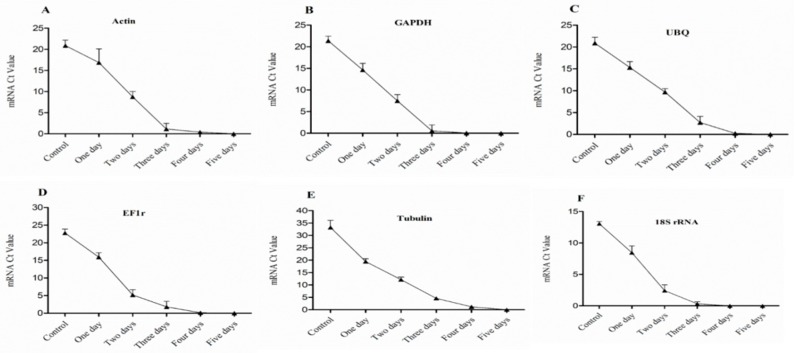
Expression level of mRNA in 3rd bud of dormant grape seedlings. (**A**) Actin, (**B**) GAPDH, (**C**) UBQ, (**D**) EF1r, (**E**) Tubulin, and (**F**) 18S rRNA were determined by qRT-PCR (1–5 days). Vertical bars represented standard error (SE) of means (*n* = 3). Different values indicated as statistical differences at *p* < 0.05 as determined by Duncan’s multiple range tests.

**Figure 9 plants-08-00577-f009:**
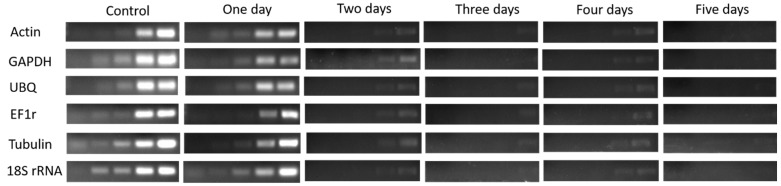
(Sq.) RT-PCR analysis of mRNA level in the 3rd bud during 1–5 days of treatment in dormant grape seedlings. 18, 21, 23, 27, and 30 cycles were used for Actin, GHAPDH, UBQ, and EF1r; 27, 30 33, 36, and 39 cycles were used for Tubulin; and 12, 15, 18, 21, and 27 cycles were used for 18S rRNA. Vertical bars represented standard error (SE) of means (*n* = 3). Different values indicated as statistical differences at *p* < 0.05 as determined by Duncan’s multiple range tests.

**Figure 10 plants-08-00577-f010:**
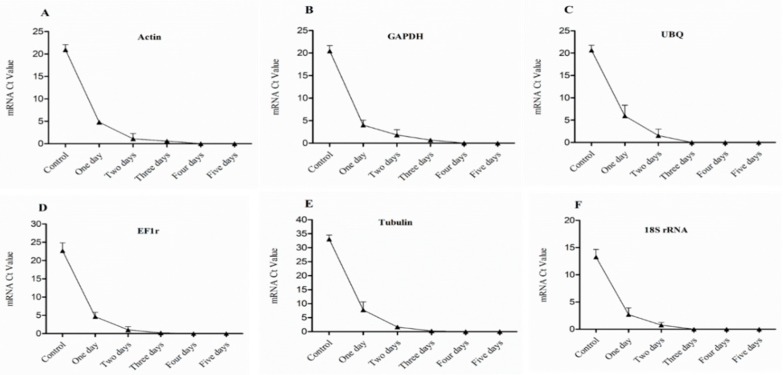
Expression level of mRNA in 4th bud of dormant grape seedlings. (**A**) Actin, (**B**) GAPDH, (**C**) UBQ, (**D**) EF1r, (**E**) Tubulin, and (**F**) 18S rRNA were determined by qRT-PCR (1–5 days). Vertical bars represented standard error (SE) of means (*n* = 3). Different values indicated as statistical differences at *p* < 0.05 as determined by Duncan’s multiple range tests.

**Figure 11 plants-08-00577-f011:**
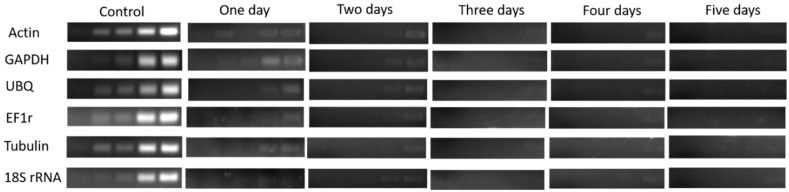
(Sq.) RT-PCR analysis of mRNA level in the 4th bud during 1–5 days of treatment in dormant grape seedlings. 18, 21, 23, 27, and 30 cycles were used for Actin, GHAPDH, UBQ, and EF1r; 27, 30 33, 36, and 39 cycles were used for Tubulin; and 12, 15, 18, 21, and 27 cycles were used for 18S rRNA. Vertical bars represented standard error (SE) of means (*n* = 3).

**Figure 12 plants-08-00577-f012:**
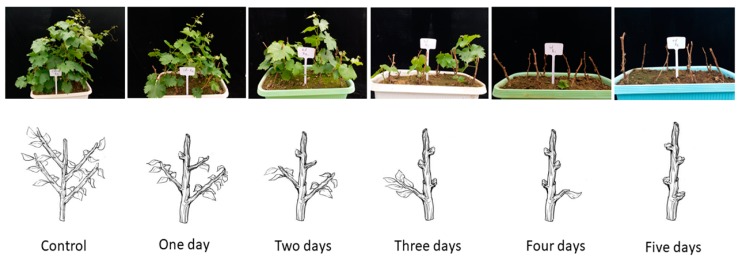
Morphological changes and viability rate of grape seedlings treated at different times under bare roots condition.

**Figure 13 plants-08-00577-f013:**
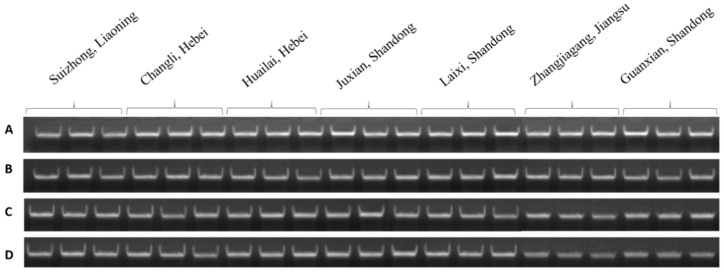
Agarose gel analysis of DNA extracted from dormant grape seedlings (**A**) 1st bud (lower bud), (**B**) 2nd bud, (**C**) 3rd bud, and (**D**) 4th bud (upper bud) tissues.

**Figure 14 plants-08-00577-f014:**
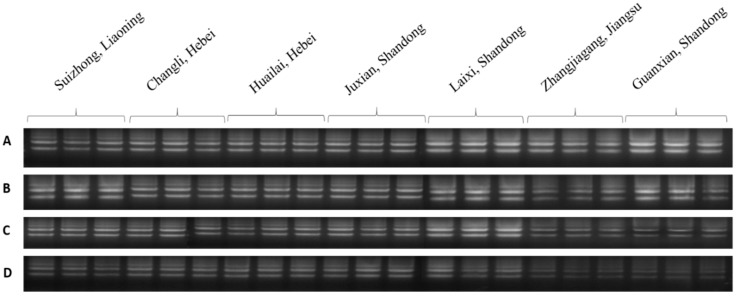
Agarose gel analysis of total RNA extracted from grape dormant seedlings (**A**) 1st bud (lower bud), (**B**) 2nd bud, (**C**) 3rd bud, and (**D**) 4th bud (upper buds) tissues.

**Figure 15 plants-08-00577-f015:**
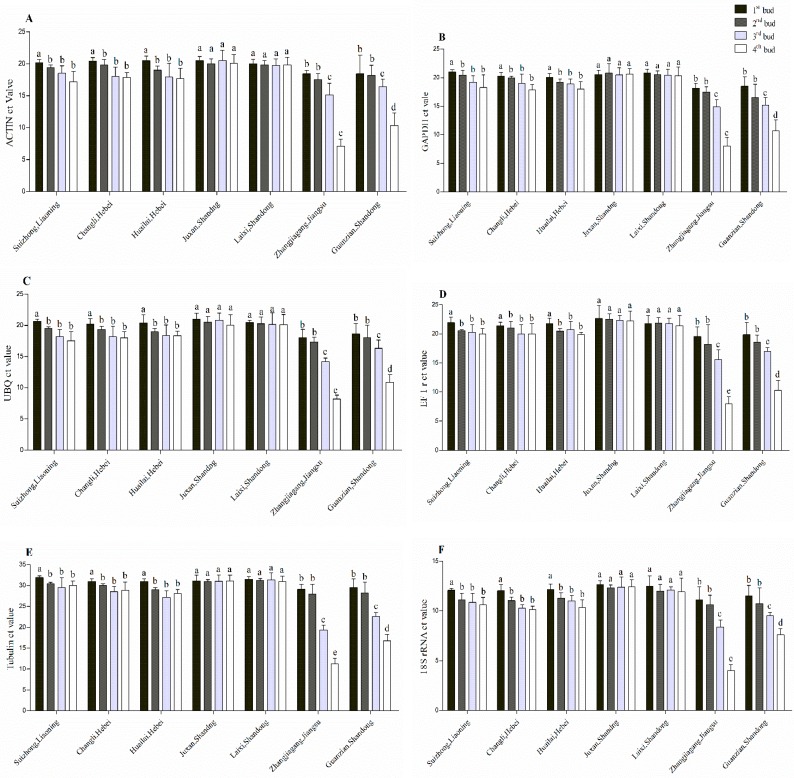
qRT-PCR analysis of mRNA level in four buds in dormant grape seedlings. (**A**) Actin, (**B**) GHAPDH, (**C**) UBQ, (**D**) EF1r, (**E**) Tubulin, and (**F**) 18S rRNA. Vertical bars represented standard error (SE) of means (*n* = 3). Different superscript letters indicated statistical differences at *p* < 0.05 as determined by Duncan’s multiple range tests.

**Figure 16 plants-08-00577-f016:**
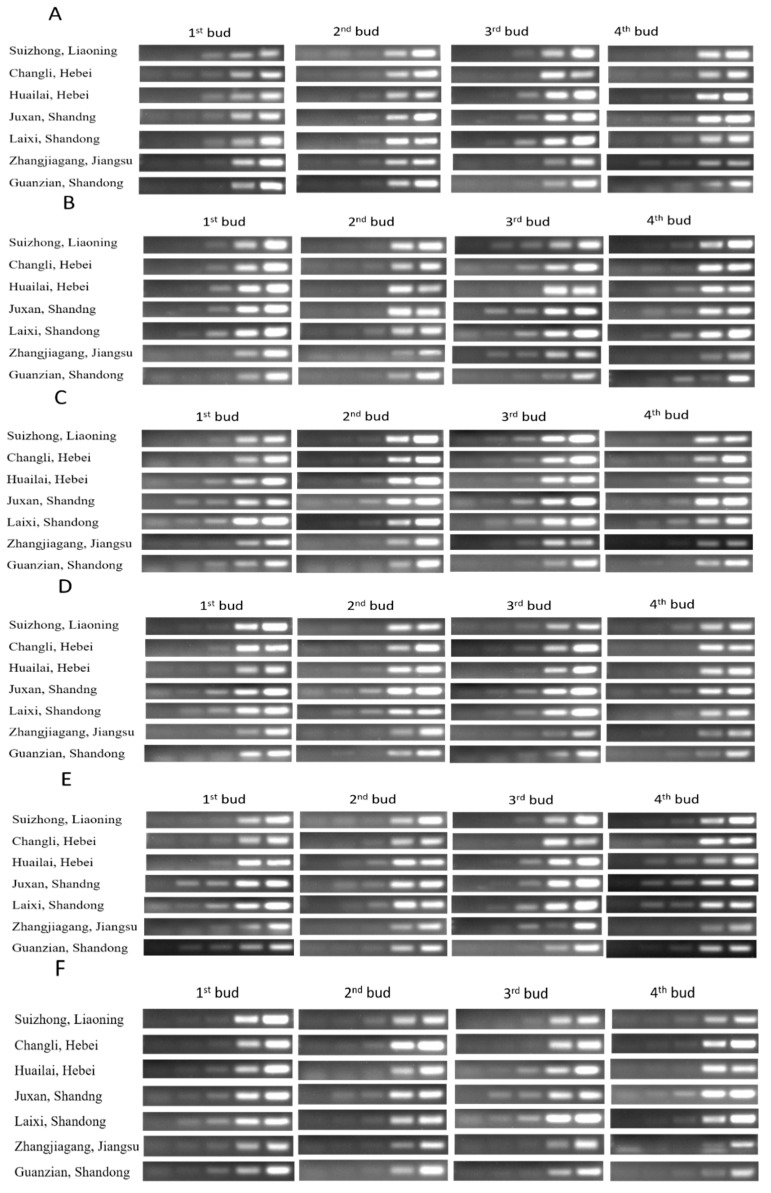
(Sq.) RT-PCR analysis of mRNA level in four buds of dormant grape seedlings produced in different geographical regions of china. (**A**) Actin, (**B**) GHAPDH, (**C**) UBQ, (**D**) EF1r, (**E**) Tubulin, and (**F**) 18S rRNA; 18, 21, 23, 27, and 30 cycles were used for Actin, GHAPDH, UBQ, and EF1r; 27, 30 33, 36, and 39 cycles were used for Tubulin; and 12, 15, 18, 21, and 27 cycles were used for 18S rRNA.

**Figure 17 plants-08-00577-f017:**
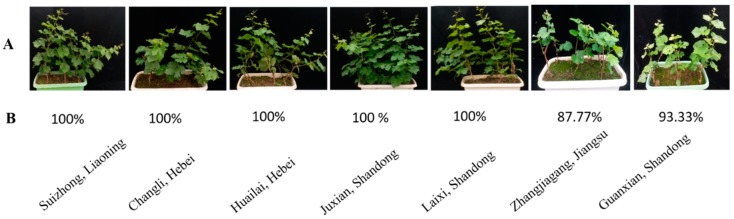
Morphological appearance and growth differences of Kyoho grape seedlings produced in seven different areas. (**A**) Growth appearance. (**B**) Survival rate.

**Figure 18 plants-08-00577-f018:**
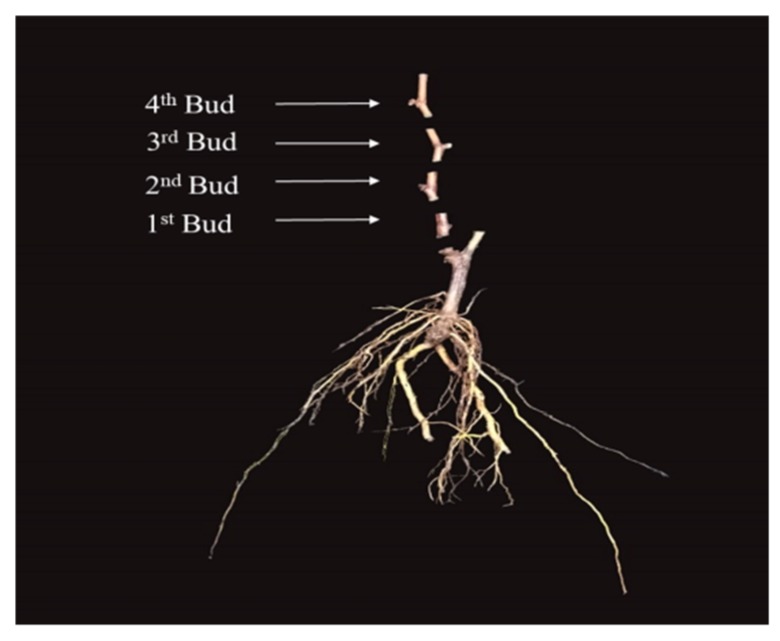
Cutting of the seedlings into 4 parts for sampling purpose.

**Table 1 plants-08-00577-t001:** UV abstraction of total DNA extracted from four different buds of the treated grape seedlings with bare roots.

DNA Quantity Analysis of Bare Roots Treated Seedlings							
Buds	OD & Concentration	Control	One Day	Two Days	Three Days	Four Days	Five Days
1st Bud	OD 260/280	1.95 ± 0.05	1.92 ± 0.04	1.82 ± 0.06	1.73 ± 0.03	1.23 ± 0.07	1.14 ± 0.03
	OD 260/230	1.96 ± 0.03	1.91 ± 0.03	1.73 ± 0.11	1.63 ± 0.07	0.93 ± 0.11	0.86 ± 0.06
	concentration(µg·mL^−1^)	1220 ± 28	1001 ± 46	891 ± 83	742 ± 27	443 ± 26	382 ± 33
2nd Bud	OD 260/280	1.93 ± 0.04	1.83 ± 0.04	1.71 ± 0.05	1.21 ± 0.03	1.2 ± 0.05	1.12 ± 0.05
	OD 260/230	1.92 ± 0.12	1.86 ± 0.03	1.62 ± 0.07	0.95 ± 0.05	0.76 ± 0.03	0.62 ± 0.02
	concentration(µg·mL^−1^)	1271 ± 53	977 ± 37	726 ± 73	419 ± 72	325 ± 25	267 ± 44
3rd Bud	OD 260/280	1.96 ± 0.07	1.74 ± 0.02	1.22 ± 0.09	1.19 ± 0.04	1.13 ± 0.03	1.07 ± 0.03
	OD 260/230	1.92 ± 0.04	1.73 ± 0.07	0.82 ± 0.03	0.72 ± 0.07	0.66 ± 0.07	0.32 ± 0.01
	concentration(µg·mL^−1^)	1239 ± 51	821 ± 33	629 ± 42	383 ± 92	298 ± 38	215 ± 23
4th Bud	OD 260/280	1.95 ± 0.1	1.18 ± 0.03	1.21 ± 0.02	1.16 ± 0.03	1.09 ± 0.04	1.05 ± 0.03
	OD 260/230	1.910.08	1.03 ± 0.02	0.76 ± 0.06	0.63 ± 0.07	0.56 ± 0.03	0.29 ± 0.01
	concentration(µg.mL^−1^)	1193 ± 45	522 ± 41	437 ± 47	314 ± 43	213 ± 42	127 ± 19

DNA Quantification using spectrophotometric measurement of UV absorption at wavelengths 230, 260, and 280 nm. DNA purity was determined by OD Values ratio OD 260: OD 280 and OD 260: OD 230 (*n* = 3). The concentration was calculated using formula, DNA concentration (µg/mL) =OD.

**Table 2 plants-08-00577-t002:** UV abstraction of total RNA extracted from four different buds of the treated grape seedlings with bare roots.

RNA Quantity Analysis of Bare Roots Treated Seedlings							
Buds	Od & Concentration	Control	One Day	Two Days	Three Days	Four Days	Five Days
1st Bud	OD 260/280	1.95 ± 0.12	1.9 ± 0.08	1.81 ± 0.09	1.72 ± 0.11	1.55 ± 0.05	1.21 ± 0.06
	OD 260/230	1.94 ± 0.04	1.52 ± 0.03	1.52 ± 0.05	1.42 ± 0.08	1.07 ± 0.07	0.9 ± 0.04
	concentration(µg·mL^−1^)	1221 ± 82	1100 ± 39	980 ± 44	833 ± 21	579 ± 23	322 ± 47
2nd Bud	OD 260/280	1.95 ± 0.11	1.82 ± 0.04	1.73 ± 0.04	1.62 ± 0.03	1.42 ± 0.02	1.18 ± 0.07
	OD 260/230	1.96 ± 0.05	1.51 ± 0.2	1.46 ± 0.07	1.03 ± 0.06	0.92 ± 0.09	0.68 ± 0.02
	concentration(µg·mL^−1^)	1253 ± 57	1053 ± 29	930 ± 64	589 ± 52	513 ± 37	282 ± 46
3rd Bud	OD 260/280	1.87 ± 0.08	1.71 ± 0.06	1.440.03	1.31 ± 0.03	1.21 ± 0.03	1.12 ± 0.03
	OD 260/230	1.92 ± 0.02	1.47 ± 0.04	0.93 ± 0.07	0.81 ± 0.04	0.68 ± 0.06	0.54 ± 0.03
	concentration(µg·mL^−1^)	1219 ± 83	970 ± 19	687 ± 53	407 ± 71	310 ± 26	219 ± 18
4th Bud	OD 260/280	1.94 ± 0.04	1.19 ± 0.05	1.18 ± 0.04	1.15 ± 0.08	1.11 ± 0.03	1.05 ± 0.08
	OD 260/230	1.97 ± 0.11	0.93 ± 0.12	0.73 ± 0.09	0.55 ± 0.03	0.41 ± 0.07	0.36 ± 0.04
	concentration(µg.mL^−1^)	1228 ± 32	622 ± 19	483 ± 42	320 ± 27	277 ± 19	175 ± 34

RNA Quantification by spectrophotometric measurement of UV absorption at wavelengths 230, 260, and 280 nm. RNA purity was determined by the OD Values ratio OD 260: OD 280 and OD 260: OD 230 (*n* = 3). The concentration was calculated using formula RNA concentration (µg/mL) =OD.

**Table 3 plants-08-00577-t003:** Relation of plant growth in different buds 10, 20, and 30 days after first bud burst and survival rate of dormant treated grape seedlings.

Growth Analysis of Four Buds and Survival Rate of the Treated Seedlings									
Treatment	Buds	Bud Burst Time (DAP)	Growth Rate (10 DAFBB)		Growth Rate (20 DAFBB)		Growth Rate (30 DAFBB)		Seedling Survival Rate (%)
			Shoots Length (cm)	Number of Leaves Per Shoot	Shoots Length (cm)	Number of Leaves per Shoot	Shoots Length (cm)	Number of Leaves Per Shoot	
Control	1st bud	21	14.90 ± 4.64 ^a^	9 ± 1.39 ^a^	22.15 ± 9.26 ^a^	17 ± 5.39 ^a^	37.30 ± 7.09 ^a^	27 ± 4.75 ^a^	100.00
	2nd bud	21	13.38 ± 5.32 ^a^	8 ± 3.41 ^a^	23.17 ± 2.14 ^a^	20 ± 2.85 ^a^	37.83 ± 6.62 ^a^	28 ± 7.63 ^a^	
	3rd bud	21	15.93 ± 6.48 ^a^	10 ± 2.95 ^a^	21.85 ± 4.73 ^a^	16 ± 2.37 ^a^	34.87 ± 3.75 ^a^	25 ± 5.73 ^a^	
	4th bud	21	14.16 ± 4.91 ^a^	9 ± 3.31 ^a^	25.74 ± 5.92 ^a^	24 ± 38 ^a^	40.51 ± 2.73 ^a^	31 ± 3.99 ^a^	
one day	1st bud	23	11.96 ± 3.29 ^b^	7 ± 2.61 ^b^	18.32 ± 2.53 ^b^	14 ± 4.72 ^b^	25.69 ± 3.71 ^b^	16 ± 5.48 ^b^	85.55
	2nd bud	24	8.13 ± 2.74 ^c^	5 ± 1.35 ^c^	15.35 ± 3.50 ^c^	10 ± 1.42 ^c^	18.74 ± 2.37 ^c^	19 ± 3.31 ^c^	
	3rd bud	25	5.61 ± 2.47 ^d^	4 ± 1.17 ^d^	10.79 ± 2.16 ^d^	5 ± 1.03 ^d^	13.76 ± 0.93 ^d^	8 ± 2.62 ^d^	
	4th bud	28	0.54 ± 0.53 ^g^	0 ± 0.21 ^g^	0.83 ± 0.96 ^g^	1 ± 0.43 ^g^	0.84 ± 0.57 ^g^	1 ± 0.53 ^g^	
two days	1st bud	28	5.63 ± 2.11 ^d^	4 ± 2.84 ^d^	9.40 ± 3.26 ^d^	6 ± 2.03 ^d^	14.35 ± 5.82 ^d^	10 ± 2.69 ^d^	75.55
	2nd bud	30	3.86 ± 1.21 ^e^	3 ± 0.94 ^e^	7.15 ± 2.04 ^e^	5 ± 1.37 ^e^	11.22 ± 2.32 ^e^	7 ± 0.98 ^e^	
	3rd bud	33	0.31 ± 0.31 ^g^	0 ± 0.20 ^g^	0.75 ± 0.84 ^g^	1 ± 0.74 ^g^	0.77 ± 0.64 ^g^	1 ± 0.43 ^g^	
	4th bud	0	0.00	0.00	0.00	0.00	0.00	0.00	
three days	1st bud	36	2.56 ± 1.78 ^f^	2 ± 1.65 ^f^	4.25 ± 2.62 ^f^	3 ± 3.41 ^f^	8.60 ± 2.43 ^f^	6 ± 1.71 ^f^	51.11
	2nd bud	39	0.00	0.00	0.74 ± 0.03 ^g^	1 ± 0.04 ^g^	0.79 ± 0.04 ^g^	1 ± 0.06 ^g^	
	3rd bud	0	0.00	0.00	0.00	0.00	0.00	0.00	
	4th bud	0	0.00	0.00	0.00	0.00	0.00	0.00	
four days	1st bud	47	0.00	0.00	0.28 ± 0.04 ^g^	1 ± 0.01 ^g^	0.42 ± 0.31 ^g^	1 ± 0.07 ^g^	2.22
	2nd bud	0	0.00	0.00	0.00	0.00	0.00	0.00	
	3rd bud	0	0.00	0.00	0.00	0.00	0.00	0.00	
	4th bud	0	0.00	0.00	0.00	0.00	0.00	0.00	
five days	1st bud	0	0.00	0.00	0.00	0.00	0.00	0.00	0.00
	2nd bud	0	0.00	0.00	0.00	0.00	0.00	0.00	
	3rd bud	0	0.00	0.00	0.00	0.00	0.00	0.00	
	4th bud	0	0.00	0.00	0.00	0.00	0.00	0.00	

Growth analysis of four buds of grape seedlings treated with bare roots condition. DAP: days after planting. DAFBB: days after first bud burst. ^a–g^ means in a row with different superscript differ significantly (*p* < 0.05). The data presented are mean± SD (*n* = 30). Means are significantly (*p* < 0.05) different within a column.

**Table 4 plants-08-00577-t004:** DNA analysis of four buds in dormant Kyoho grapes produced in different regions.

DNA Analysis of Four Buds in Different Regions’ Seedlings				
Region of Seedlings Production		Optical Density OD260/OD280	Optical Density OD260/OD230	Concentration (µg.mL^−1^)
Suizhong, Liaoning	1st bud	1.94 ± 0.01	1.92 ± 0.02	1222 ± 36
	2nd bud	1.87 ± 0.03	1.86 ± 0.03	1159 ± 47
	3rd bud	1.88 ± 0.02	1.83 ± 0.04	1173 ± 68
	4th bud	1.84 ± 0.04	1.84 ± 0.02	1187 ± 56
Changli, Hebei	1st bud	1.92 ± 0.05	1.91 ± 0.03	1217 ± 61
	2nd bud	1.86 ± 0.02	1.87 ± 0.07	1074 ± 47
	3rd bud	1.87 ± 0.03	1.83 ± 0.04	1095 ± 56
	4th bud	1.85 ± 0.04	1.81 ± 0.03	1166 ± 37
Huailai, Hebei	1st bud	1.97 ± 0.05	1.92 ± 0.02	1213 ± 91
	2nd bud	1.87 ± 0.02	1.84 ± 0.02	1062 ± 62
	3rd bud	1.89 ± 0.01	1.85 ± 0.03	1027 ± 26
	4th bud	1.84 ± 0.05	1.84 ± 0.04	1193 ± 56
Juxian, Shandong	1st bud	1.96 ± 0.03	1.95 ± 0.02	1351 ± 37
	2nd bud	1.94 ± 0.04	1.91 ± 0.06	1218 ± 42
	3rd bud	1.94 ± 0.02	1.90 ± 0.03	1302 ± 37
	4th bud	1.93 ± 0.03	1.97 ± 0.02	1344 ± 39
Laixi, Shandong	1st bud	1.92 ± 0.02	1.92 ± 0.02	1310 ± 39
	2nd bud	1.91 ± 0.06	1.95 ± 0.02	1206 ± 62
	3rd bud	1.93 ± 0.02	1.93 ± 0.07	1295 ± 37
	4th bud	1.89 ± 0.02	1.92 ± 0.03	1294 ± 81
Zhangjiagang, Jiangsu	1st bud	1.82 ± 0.13	1.81 ± 0.06	911 ± 54
	2nd bud	1.71 ± 0.08	1.69 ± 0.01	798 ± 36
	3rd bud	1.63 ± 0.03	1.62 ± 0.03	702 ± 24
	4th bud	1.142 ± 0.06	1.39 ± 0.05	617 ± 44
Guanxian, Shandong	1st bud	1.84 ± 0.04	1.84 ± 0.06	915 ± 93
	2nd bud	1.73 ± 0.02	1.77 ± 0.04	812 ± 72
	3rd bud	1.67 ± 0.03	1.71 ± 0.01	738 ± 42
	4th bud	1.46 ± 0.06	1.40 ± 0.04	658 ± 46

DNA Quantification by spectrophotometric measurement of UV absorption at wavelengths 230, 260, and 280 nm. DNA purity was determined by the OD Values ratio OD 260: OD 280 and OD 260: OD 230 (*n* = 3). The concentration was calculated using formula DNA concentration (µg/mL) =OD.

**Table 5 plants-08-00577-t005:** RNA analysis of four buds in dormant Kyoho grapes produced in different regions.

RNA Analysis of Four Buds in Different Regions’ Seedlings				
Region of Seedlings Production		Optical Density OD260/OD280	Optical Density OD260/OD230	Concentration (µg·mL^−1^)
Suizhong, Liaoning	1st bud	1.93 ± 0.02	1.92 ± 0.03	1160 ± 32
	2nd bud	1.9 ± 0.6	1.91 ± 0.04	110 ± 41
	3rd bud	1.87 ± 0.07	1.88 ± 0.07	1093 ± 27
	4th bud	1.84 ± 0.01	1.84 ± 0.01	1131 ± 24
Changli, Hebei	1st bud	1.92 ± 0.05	1.91 ± 0.03	1095 ± 33
	2nd bud	1.87 ± 0.06	1.91 ± 0.04	1135 ± 29
	3rd bud	1.86 ± 0.05	1.87 ± 0.02	1063 ± 34
	4th bud	1.86 ± 0.02	1.86 ± 0.02	1092 ± 63
Huailai, Hebei	1st bud	1.91 ± 0.21	1.90 ± 0.01	1157 ± 66
	2nd bud	1.88 ± 0.07	1.86 ± 0.02	1021 ± 38
	3rd bud	1.92 ± 0.1	1.85 ± 0.04	1023 ± 26
	4th bud	1.86 ± 0.03	1.86 ± 0.02	1123 ± 18
Juxian, Shandong	1st bud	1.97 ± 0.02	1.95 ± 0.03	1295 ± 23
	2nd bud	1.96 ± 0.05	1.93 ± 0.02	1219 ± 41
	3rd bud	1.92 ± 0.03	1.91 ± 0.05	1257 ± 37
	4th bud	1.97 ± 0.03	1.97 ± 0.02	1220 ± 26
Laixi, Shandong	1st bud	1.94 ± 0.01	1.95 ± 0.03	1225 ± 82
	2nd bud	1.96 ± 0.04	1.94 ± 0.02	1252 ± 39
	3rd bud	1.98 ± 0.05	1.94 ± 0.07	1283 ± 41
	4th bud	1.94 ± 0.02	1.94 ± 0.03	1151 ± 38
Zhangjiagang, Jiangsu	1st bud	1.81 ± 0.08	1.80 ± 0.04	924 ± 25
	2nd bud	1.72 ± 0.03	1.71 ± 0.08	847 ± 46
	3rd bud	1.67 ± 0.07	1.64 ± 0.05	783 ± 57
	4th bud	1.41 ± 0.05	1.33 ± 0.05	683 ± 29
Guanxian, Shandong	1st bud	1.83 ± 0.07	1.81 ± 0.04	936 ± 87
	2nd bud	1.73 ± 0.04	1.69 ± 0.03	866 ± 49
	3rd bud	1.72 ± 0.01	1.67 ± 0.01	791 ± 27
	4th bud	1.45 ± 0.06	1.37 ± 0.06	703 ± 45

RNA Quantification using the spectrophotometric measurement of UV absorption at wavelengths 230, 260, and 280 nm. RNA purity was determined by the OD Values ratio OD 260: OD 280 and OD 260: OD 230 (*n* = 3). The concentration was calculated using formula RNA concentration (µg/mL) =OD.

**Table 6 plants-08-00577-t006:** Growth observation and survival rate of grape seedlings produced in seven different geographical regions of China.

Geographical Regions of the Seedlings	Buds	BBD (DAP)	Growth Rate 10 (DAFBB)		Growth Rate 20 (DAFBB)		Growth Rate 30 (DAFBB)		Survival Rate (%)
			Shoots Length (cm)	Number of Leaves Per Plant	Shoots Length (cm)	Number of Leaves Per Plant	Shoots Length (cm)	Number of Leaves Per Plant	
Suizhong, Liaoning	1st	18 March 2019	17.44 ± 3.12 ^a^	8 ± 2.20 ^a^	23.33 ± 5.32 ^a^	12 ± 2.64 ^a^	43.88 ± 5.81 ^a^	33 ± 4.28 ^a^	100.00
	2nd		15.32 ± 3.42 ^b^	8 ± 3.21 ^b^	21.85 ± 4.63 ^b^	12 ± 3.42 ^b^	40.34 ± 4.42 ^b^	30 ± 4.32 ^b^	
	3rd		15.21 ± 4.37 ^b^	7 ± 4.53 ^b^	20.86 ± 4.87 ^b^	11 ± 2.24 ^b^	40.84 ± 3.72 ^b^	27 ± 7.37 ^b^	
	4th		14.39 ± 6.72 ^b^	7 ± 5.63 ^b^	20.01 ± 7.43 ^b^	10 ± 2.12 ^b^	38.99 ± 5.38 ^b^	26 ± 6.73 ^b^	
Changli, Hebei	1st	17 March 2019	17.45 ± 2.93 ^a^	10 ± 2.02 ^a^	24.54 ± 6.59 ^a^	12 ± 1.42 ^a^	43.90 ± 5.97 ^a^	34 ± 7.03 ^a^	100.00
	2nd		15.32 ± 4.32 ^b^	9 ± 2.31 ^b^	22.37 ± 3.48 ^b^	11 ± 3.28 ^b^	41.02 ± 7.57 ^b^	29 ± 5.69 ^b^	
	3rd		14.35 ± 2.21 ^b^	8 ± 2.83 ^b^	22.10 ± 0.41 ^b^	10 ± 1.12 ^b^	39.34 ± 8.73 ^b^	31 ± 2.49 ^b^	
	4th		13.23 ± 2.82 ^b^	8 ± 2.74 ^b^	21.72 ± 3.74 ^b^	10 ± 2.00 ^b^	38.95 ± 6.74 ^b^	30 ± 7.38 ^b^	
Huailai, Hebei	1st	17 March 2019	17.30 ± 2.40 ^a^	9 ± 1.39 ^a^	24.60 ± 7.41 ^a^	12 ± 3.16 ^a^	44.50 ± 3.34 ^a^	35 ± 5.94 ^a^	100.00
	2nd		16.28 ± 3.74 ^b^	9 ± 1.75 ^b^	22.15 ± 6.37 ^b^	10 ± 3.27 ^b^	40.46 ± 7.43 ^b^	35 ± 3.86 ^b^	
	3rd		16.74 ± 3.38 ^b^	8 ± 2.37 ^b^	21.74 ± 4.37 ^b^	10 ± 2.91 ^b^	42.85 ± 3.65 ^b^	32 ± 2.75 ^b^	
	4th		14.72 ± 3.17 ^b^	8 ± 4.65 ^b^	20.20 ± 2.45 ^b^	10 ± 1.69 ^b^	39.26 ± 2.74 ^b^	31 ± 5.14 ^b^	
Juxian, Shandong	1st	12 March 2019	18.32 ± 3.44 ^a^	9 ± 2.22 ^a^	25.70 ± 7.63 ^a^	13 ± 1.50 ^a^	45.71 ± 7.04 ^a^	35 ± 6.48 ^a^	100.00
	2nd		17.92 ± 2.64 ^a^	9 ± 2.12 ^a^	24.84 ± 5.38 ^a^	11 ± 3.82 ^a^	44.52 ± 6.62 ^a^	33 ± 5.43 ^a^	
	3rd		18.11 ± 3.93 ^a^	8 ± 3.25 ^a^	26.43 ± 6.72 ^a^	12 ± 4.62 ^a^	43.19 ± 3.29 ^a^	37 ± 4.23 ^a^	
	4th		18.06 ± 2.19 ^a^	9 ± 2.51 ^a^	25.73 ± 5.62 ^a^	13 ± 2.76 ^a^	45.13 ± 3.51 ^a^	34 ± 3.39 ^a^	
Laixi, Shandong	1st	14 March 2019	18.70 ± 4.64 ^a^	9 ± 1.84 ^a^	25.10 ± 5.24 ^a^	13 ± 6.39 ^a^	44.30 ± 8.09 ^a^	34 ± 4.75 ^a^	100.00
	2nd		17.97 ± 3.23 ^a^	10 ± 3.32 ^a^	26.35 ± 3.87 ^a^	14 ± 3.23 ^a^	46.42 ± 4.51 ^a^	33 ± 6.34 ^a^	
	3rd		18.17 ± 4.32 ^a^	9 ± 3.46 ^a^	24.32 ± 3.23 ^a^	15 ± 4.34 ^a^	43.29 ± 4.51 ^a^	31 ± 4.61 ^a^	
	4th		18.35 ± 3.97 ^a^	10 ± 3.28 ^a^	24.47 ± 4.51 ^a^	13 ± 2.34 ^a^	42.43 ± 2.36 ^a^	30 ± 3.42 ^a^	
Zhangjiagang, Jiangsu	1st	23 March 2019	15.40 ± 0.96 ^b^	7 ± 0.84 ^b^	22.40 ± 7.33 ^b^	10 ± 4.68 ^b^	37.21 ± 4.02 ^b^	27 ± 3.72 ^b^	87.77
	2nd		12.97 ± 1.84 ^b^	6 ± 1.14 ^b^	18.63 ± 2.12 ^b^	7 ± 2.32 ^b^	29.97 ± 4.42 ^b^	20 ± 5.43 ^b^	
	3rd		10.26 ± 0.34 ^c^	5 ± 0.92 ^c^	15.87 ± 2.31 ^c^	5 ± 1.46 ^c^	23.96 ± 3.41 ^c^	15 ± 2.74 ^c^	
	4th		8.37 ± 0.75 ^d^	4 ± 0.28 ^d^	12.27 ± 1.03 ^d^	4 ± 0.43 ^d^	18.84 ± 2.15 ^d^	11 ± 1.63 ^d^	
Guanxian, Shandong	1st	20 March 2019	15.54 ± 0.84 ^b^	8 ± 0.73 ^b^	22.80 ± 6.19 ^b^	11 ± 1.33 ^b^	38.34 ± 3.72 ^b^	29 ± 4.11 ^b^	93.33
	2nd		14.37 ± 1.20 ^b^	7 ± 0.37 ^b^	19.54 ± 3.58 ^b^	8 ± 1.81 ^b^	33.63 ± 5.63 ^b^	24 ± 3.17 ^b^	
	3rd		9.27 ± 0.53 ^c^	4 ± 0.48 ^c^	13.77 ± 1.85 ^c^	5 ± 0.45 ^c^	21.97 ± 2.23 ^c^	13 ± 1.96 ^c^	
	4th		5.96 ± 0.74 ^e^	3 ± 2.97 ^e^	9.14 ± 0.77 ^e^	3 ± 0.43 ^e^	16.88 ± 0.71 ^e^	9 ± 0.95 ^e^	

Growth performance and survival rate of the grape seedlings produced in seven different geographical regions of China. DAFBB: days after first bud burst. BBD (DAP): bud burst date (days after planting). ^a–e^ means in a row with different superscript differ significantly (*p* < 0.05). The data presented are mean± SD (*n* = 30). Means followed by different letters within a column is significantly (*p* < 0.05) different.

**Table 7 plants-08-00577-t007:** Primers used in (Sq.) RT-PCR and qRT-PCR.

Name	Sequence	Accession No	Product Size (bp)
ACTIN-F	CTTGCATCCCTCAGCACCTT	XM_002282480.4	82
ACTIN-R	TCCTGTGGACAATGGATGGA		
GAPDH-F	TTCCGTGTTCCTACTGTTG	XM_002263109.3	106
GAPDH-R	CCTCTGACTCCTCCTTGAT		
UBQ-F	GTGGTATTATTGAGCCATCCTT	XM_002273532.2	182
UBQ-R	AACCTCCAATCCAGTCATCTAC		
EF1r-F	CAAGAGAAACCATCCCTAGCTG	XM_002264364.4	92
EF1r-R	TCAATCTGTCTAGGAAAGGAAG		
Tubulin-F	TTTGCTCCACTCACTTCC	XM_002281253.4	181
Tubulin-R	TCTGCTCGTCCACTTCTT		
18S rRNA-F	TGGCCTTCGGGATCGGAGTAA	GQ849399.1	209
18S rRNA-R	ATCCCTGGTCGGCATCGTTTAT

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
