# Peer review of "Molecular Evaluation of Vitality and Survival Rate of Dormant Kyoho Grape Seedlings: A Step toward Molecular Farming"

_plants, 2019, doi:10.3390/plants8120577_

Round 1

Reviewer 1 Report

The study focuses on the impact of environmental stress on DNA quality and housekeeping gene expression in grape cuttings. The main conclusion appears to be that stress reduces both of these factors and the authors suggest that this finding could provide a valuable mechanism for assessing the viability of cutting grown plants prior to planting. Their data does convincingly demonstrate the loss of DNA and RNA quantity and quality following dehydration stress, which supports their conclusion, although I suspect that the cost of this type of analysis would act against its adoption by industry. The manuscript needs considerable editing for the quality of the English. It was some time before I realized that they were not using ‘seedling’ material, as stated, but rather cutting grown plants. Also, their choice of a dehydration stress was not explained in the introduction and it was not clear whether the material from different regions was propagated in their facility or imported rooted. This would have an impact on the conclusions. For instance was the poor performance of material from two regions due to poor field material or poor transport to their research site. Similarly, the apparent loss of DNA and RNA quality down the stem was not explained fully. In conclusion, I believe that the data is interesting and clearly a lot of work went into producing it. The standard of the manuscript, however, does not reflect this.

Author Response

Response to Reviewer 1 Comments

Ref.: Ms. No. plants-641523
" Molecular evaluation of vitality and survival rate of dormant Kyoho grape seedlings: a step toward molecular farming ". Journal of plants- MDPI

Dear reviewer!

Thank you very much for giving me opportunity to revise our MS. According to your valuable comments/suggestions, we cordially appreciate your input for revision. Please find the revised sections in red color in main MS for your information.  

Point 1:  The manuscript needs considerable editing for the quality of the English.

Response 1: We have revised the manuscript and made considerable English editing. If still required further editing, we may refer to English editing department of the journal.

Point 2:  It was some time before I realized that they were not using ‘seedling’ material, as stated, but rather cutting grown plants.

Response 2:  We have used cutting grown seedlings in this experiment produced in respective 07 geographical regions of china for equal results of the vitality and survival rate. Seedling materials related information is written in the manuscript in line (457-494).

Point 3:  Their choice of a dehydration stress was not explained in the introduction and it was not clear whether the material from different regions was propagated in their facility or imported rooted. This would have an impact on the conclusions.

Response 3: a- We modified the introduction section with adding dehydration stress information according to your suggestions (from line 55 to 61). B- Nanjing Agricultural University research site is using standard protocol for transportation of the dormant seedlings for researches. C- The facilities and growing conditions are differ in different seedlings’ producing companies Therefore, our main aim behind this research was to investigate the possible application of molecular evaluation for identification of best source of seedlings among selected companies/regions. These dormant seedlings had same morphological specifications but we cannot distinguish expected growth and survival rate in next growing season due to dormancy.

Point 4:  For instance was the poor performance of material from two regions due to poor field material or poor transport to their research site.

Response 4:  We have standard transportation system for seedlings in our university, the field materials and growing conditions were same in the field as written in the materials and methods section (line 491-494).

Point 5: Similarly, the apparent loss of DNA and RNA quality down the stem was not explained fully.

Response 5:  for analysis of DNA and RNA quality, we focused on buds because bud is the most appropriate organ for gene expression analysis in dormant seedlings and it is the main part that can burst and grow. Additionally, bud is the most sensitive part of the seedlings against dehydration reported by  (Pacey-Miller, Scott et al. 2003, CARMONA, Reginato et al. 2016).

Point 6: In conclusion, I believe that the data is interesting and clearly a lot of work went into producing it. The standard of the manuscript, however, does not reflect this.

Response 6:  As per your valuable concerns, we tried to improve and revise the manuscript in order to clearly reflect our work. Please see the revisions in main manuscript with red.

References:

CARMONA, J. R., G. Reginato and C. Peppi (2016). "Effect of dehydration during storage on viability of dormant grafted grape." Journal of the American Pomological Society 70(1): 16-25.

Pacey-Miller, T., K. Scott, E. Ablett, S. Tingey, A. Ching and R. Henry (2003). "Genes associated with the end of dormancy in grapes." Functional & integrative genomics 3(4): 144-152.

Reviewer 2 Report

The relevance of this manuscript is not in doubt.

However, I pay attention to the poor quality of the drawings and blurred captions to them. Perhaps this is due to a resolution that is less than 300 dpi.

Surprisingly, the M&M section is after the discussion section.

It is surprising to have a certain number of spelling errors in words, especially in the conclusions, which can be avoided by elementary spell checking in a text editor.

The term "vitality", which appears in the title, alternates throughout the manuscript with the term "viability". Articles (https://link.springer.com/chapter/10.1007%2F978-3-319-69126-8_8) give grounds to recommend the authors to establish clear terminological boundaries between terms.

Throughout the text of the manuscript there are small careless inaccuracies in the design, which must be carefully adjusted in accordance with the requirements.

See the comments in the attached file.

Author Response

Response to Reviewer 2 Comments

Ref.: Ms. No. plants-641523
" Molecular evaluation of vitality and survival rate of dormant Kyoho grape seedlings: a step toward molecular farming ". Journal of plants- MDPI

Dear reviewer!

Thank you very much for giving me the opportunity to revise our MS. According to your valuable comments/suggestions, we cordially appreciate your input for revision. Please find the revised sections in red color in main MS for your information.

Point 1: The names should be same for all universities.

Response 1: We revised the affiliations according to your instruction (mentioned in line 10, 12).

Point 2: The initials of the author must be placed in parentheses after each email address, for example, [email protected] (M. N).

Response 2: We revised according to your instruction (mentioned in line 8, 9, 11, 13, and 14).

Point 3: I recommend adding a sentence that answers the questions: “who are the intended readers? And why they are interested in your study?”

Response 3:  The intended readers are viticulturists and we modified the main manuscript with further considerations into your valuable comments, please see line (30, 31, 85-87).

Point 4: Latin names of species should be designed in accordance with the rules for authors.

Response 4: Referring to the journal guidelines, we revised the Latin names (line 21). 

Point 5: The decimal point must be followed by the same number of characters for all values.

Response 5: We modified according to the instruction (line 28, 29).

Point 6: It would be interesting to see at this point in the abstract the expression levels for each of the survival rate.

Response 6: Considering the journal guidelines, we would not able to exceed the abstract size more than 200 words limitation. Therefore, we concisely wrote the abstract in order to meet the journal requirements. However, we mentioned the expression level relation to the vitality and survival rate shortly in abstract (line 23-25).

Point 7: This is a fairly general and non-specific conclusion. Break the sentence into two parts that answer the questions: what do the results mean in theory? What do the results mean in practice?

Response 7: We modified the conclusion section considering your valuable comments. Please see the line 29-31 in red.

Point 8: (CM), Here and everywhere in the text should be lower case letters.

Response 8: Yes sir. You are right. We revised accordingly.

Point 9: Why the average number of leaves has two decimal places, this goes against the physical meaning of the number of leaves, which is always integer.

Response 9: Alright, we revised according to your instruction and write just the integer numbers and ignore the decimal places (highlighted in table 3 & 6).

Point 10: Here and everywhere in the text, specify the sample size for each dataset.

Response 10: We have specified the sample size in the legend of the tables and figures where is needed. (Line 257, 382).

Point 11:  here and everywhere in the text, how do you measure length an ordinary ruler can provide measurement accuracy up to one decimal place and you have two digits, measuring with more accurate tools significantly increase the cost of the process. Please explain in detail.

Response 11: We measured the length of shoot (n=30) with ordinary ruler and then we used IBM SPSS statistics, version 19 software (SPSS Inc., Chicago, IL, USA) for confirmation of the data. The data was presented mean ± SD. The two decimal digits presented for accurate results and confirmation of molecular data by field performance.

Point 12: Explain in more details the order of random selection of seedlings.

Response 12: completely randomized design was used for randomly selection (mentioned in line 462-464).

Point 13: At the expense of what the effect of seed of a molecular technique declared in the abstract turns out?

Response 13: Previously, the vitality and survival rate of grape seedlings was evaluated by long growing trial, and the replacement of the death plants in the field was time consuming and costly. Thus, we applied molecular technique to estimate vitality and survival rate of dormant grape seedlings rapidly and accurately. It can support viticulturists in selection of potential seedlings and avoid time consumption.   

Point 14: Traditionally, the altitude above sea level is also indicated.

Response 14: we have mentioned the elevation from sea level in the text (line 477-482).

Point 15: What tools was used to make the cut, at what angle? Specify the range in millimeters for the distance from the fourth bud to the cut line. How did you control that distance, by eye.

Response 15: the upper parts of the seedlings (25mm above from 4th bud using ordinary ruler commencing ground level) were cut in a horizontal angle direction by pruning scissor for growth comparison (written in line 487-489).

Point 16: Describe in details the technique of collecting from the kidney from extraction to freezing, what tools and equipment were used?

Response 16: The bud samples were collected from four parts of dormant seedlings as shown in (Figure 18) and then immediately frozen in liquid nitrogen and stored in -80°C up to RNA and RNA extraction. Each part containing one bud, buds of each part were collected for qRT-PCR, (Sq.) RT-PCR analysis and DNA and RNA quantification and quality analysis.

Point 17: How and with what equipment these photos are were obtained. For what purpose are the two cuttings presented here?

Response 17: the photo was obtained by ordinary camera to clarify the method; we have removed the second repeated cutting picture according to your instruction.

Point 18: How long does it take to analyze one sample? Have you used a standard technique or improved performance? Describe the advantages of your method. Specify the equipment and tools used.

Response 19: Around two hours were needed to analyze one sample; Genomic DNA was isolated according to the standard protocol of Vazyme FastPure Plant DNA Isolation Mini Kit (Vazyme Biotech Co., Ltd.). This kit is specially designed for plant samples with high content of polysaccharides and polyphenols, the kit uses silica gel membrane purification technology and new unique solution system. No toxic reagents such us phenol chloroform are needed in this kit and no time consuming alcohol precipitation is required during the isolation process. The extraction method is described in (line 504-520).

Point 20: About the selection of the bark above (4. 1. 2.) is not mentioned.

Response 20: We have done several pre-experiments prior to our real study. The term bark was mistakenly written in manuscript, now we revised accordingly.

Point 21: How long does it take to analyze one sample? Have you used a standard technique or improved it and improved performance? Describe the advantages of your method. Specify the equipment and tools used.

Response 21: Analysis of one sample takes less than two hours, Total RNA was isolated according to the protocol of the E.Z.N.A. ®Plant RNA Kit. No. R6827-02. Kit (OMEGA bio-tek Co., Ltd.). We used standard protocol with some improvement advised by the producing company for removing the polysaccharides and polyphenols content in the sample, (2-mercaproethanol was added to RCL buffer 1µL per ml). The advantages of this method: in certain plant samples, RNA isolation is difficult due to their large amount of polysaccharides and phenolic compounds; this protocol involves a simple and rapid precipitation that will remove much of these compounds.

Point 22: For such large-scale study, correct statistical processing of data is an important part. Therefore, this section should be expended and described in more detail. What methods were used to assess the morality of the samples? What number of measurements (repetitions) N was taken for each date set? Why Duncan test? And so on.

Response 22: The mentioned section expanded and describes in details the statistical analysis part;

We used one-way analysis of variance (ANOVA) method and then followed by the Duncan post hoc test to assess the morality of sample in minor significances. The number of measurements for the repetitions was 3 for RNA and DNA quality analysis and expression, for growth analysis the repetitions were 30 in number. We used Duncan test to detect high and accurate significances among the samples. 

Point 23: What does this word mean?

Response 23: we have made grammatical and spelling improvement in conclusion part.  

Point 24: This sentence is allegorical and is used more often in popular science texts. Explain more specifically what results will increase the profit of viticulture and will be interest to owners of vineyards. The “window” may not open for other growing conditions and other varieties of grapes. 

Response 24: We made the required improvement in conclusion part considering your suggestions. please read the revision in red (line 586-593).
